# Neuroblastoma Tumor-Associated Mesenchymal Stromal Cells Regulate the Cytolytic Functions of NK Cells

**DOI:** 10.3390/cancers15010019

**Published:** 2022-12-20

**Authors:** Sabina Di Matteo, Maria Antonietta Avanzini, Gloria Pelizzo, Valeria Calcaterra, Stefania Croce, Grazia Maria Spaggiari, Charles Theuer, Gianvincenzo Zuccotti, Lorenzo Moretta, Andrea Pelosi, Bruno Azzarone

**Affiliations:** 1Tumor Immunology Unit, Bambino Gesù Children’s Hospital, IRCCS, 00146 Rome, Italy; 2Immunology and Transplantation Laboratory, Pediatric Hematology Oncology Unit, Fondazione IRCCS Policlinico San Matteo, 27100 Pavia, Italy; 3Department of Biomedical and Clinical Science, University of Milan, 20122 Milan, Italy; 4Pediatric Surgery Department, “Vittore Buzzi” Children’s Hospital, 20154 Milan, Italy; 5Department of Internal Medicine, University of Pavia, 27100 Pavia, Italy; 6Pediatric Department, “Vittore Buzzi” Children’s Hospital, 20154 Milan, Italy; 7Department of Experimental Medicine (DIMES), University of Genoa, 16126 Genoa, Italy; 8Laboratory of Clinical and Experimental Immunology, IRCCS Istituto Giannina Gaslini, 16147 Genoa, Italy; 9Tracon Pharmaceuticals, Inc., San Diego, CA 92122, USA

**Keywords:** neuroblastoma, tumor-associated mesenchymal stromal cells (TA-MSC), WWTR1, TAZ, GD2, natural killer cells, tumor microenvironment, immunosuppression, senescence, CAF

## Abstract

**Simple Summary:**

Different components of the tumor microenvironment, such as cancer-associated fibroblasts and tumoral mesenchymal stromal cells, play a major role in cancer progression, metastatic spread and resistance to chemo-immunotherapy. Neuroblastoma tumor-associated mesenchymal stromal cells (NB-TA-MSC) have been extensively characterized for their pro-tumorigenic properties, while their immunosuppressive potential, especially against NK cells, has not been investigated. Herein, we show for the first time that primary young/proliferating NB-TA-MSC, but not senescent ones, are resistant to direct cell lysis by activated NK cells, inhibiting proliferation, cytolytic activity and functional markers of freshly isolated NK cells. NB-TA-MSC express the neuroblastoma marker GD2, the most common target for NB immunotherapy; thus, they represent priority targets whose elimination is essential for improved NB immunotherapy. From a future perspective, in vivo eradication or disabling of NB-TA-MSC could be achieved using monoclonal antibodies directed against targets different from GD2 or inducing their senescence.

**Abstract:**

Neuroblastoma tumor-associated mesenchymal stromal cells (NB-TA-MSC) have been extensively characterized for their pro-tumorigenic properties, while their immunosuppressive potential, especially against NK cells, has not been thoroughly investigated. Herein, we study the immune-regulatory potential of six primary young and senescent NB-TA-MSC on NK cell function. Young cells display a phenotype (CD105+/CD90+/CD73+/CD29+/CD146+) typical of MSC cells and, in addition, express high levels of immunomodulatory molecules (MHC-I, PDL-1 and PDL-2 and transcriptional-co-activator WWTR1), able to hinder NK cell activity. Notably, four of them express the neuroblastoma marker GD2, the most common target for NB immunotherapy. From a functional point of view, young NB-TA-MSC, contrary to the senescent ones, are resistant to activated NK cell-mediated lysis, but this behavior is overcome using anti-CD105 antibody TRC105 that activates antibody-dependent cell-mediated cytotoxicity. In addition, proliferating NB-TA-MSC, but not the senescent ones, after six days of co-culture, inhibit proliferation, expression of activating receptors and cytolytic activity of freshly isolated NK. Inhibitors of the soluble immunosuppressive factors L-kynurenine and prostaglandin E2 efficiently counteract this latter effect. Our data highlight the presence of phenotypically heterogeneous NB-TA-MSC displaying potent immunoregulatory properties towards NK cells, whose inhibition could be mandatory to improve the antitumor efficacy of targeted immunotherapy.

## 1. Introduction

Neuroblastoma (NB), the most common extracranial solid tumor in children, accounting for 10% to 20% of deaths of pediatric malignancies, arises from the embryonic neural crest cells and is characterized by an early age of onset, frequent metastases and rare cases of spontaneous regression [1].

Despite chemotherapy and immunotherapy approaches, approximately 40% of patients with NB still relapse [2,3]. Indeed, mechanisms of immunosuppression and immune evasion hinder therapies based on the employment of disialoganglioside GD2 monoclonal antibodies, GD2 chimeric antigen receptor (CAR) T-cells, and natural killer (NK) cells [2,3,4]. A growing body of literature has highlighted the importance of the tumor microenvironment (TME) in cancer progression, metastatic spread, and resistance to chemotherapy and immunotherapy [5,6,7,8,9,10]. Several components of the TME, including cancer-associated fibroblasts (CAF), tumor-derived mesenchymal stem cells (T-MSC), myeloid-derived suppressor cells (MDSC), and tumor-associated-macrophages (TAM) compromise immune effector functions against tumor cells [11,12,13,14,15,16,17,18]. Primary tumor-associated mesenchymal stromal cells (TA-MSC) have been isolated from NB. They share phenotypic and functional characteristics with bone marrow–derived MSCs [19,20,21,22,23,24,25], whereas they may display a heterogeneous expression of GD2 [19,24]. Indeed, Borriello and collaborators expanded NB-TA-MSC- isolated through GD2 negative isolation; thus, these primary cultures were GD2 negative [19], while Wu and collaborators isolated NB-TA-MSC without a negative selection for GD2, and they essentially expanded GD2^+^ NB-TA-MSC [24].

Interestingly, the interplay between NK cells and MSC is an exciting field of investigation for their potential clinical applications [26]. NB-TA-MSC regulate T cells [20], but their effects on NK cells have not been demonstrated, despite the fact that NB with low NK cell density are also devoid of intratumoral dendritic cells and have a poor prognosis [18]. Since NK cells are key components of the innate immune system that mediates the lysis of cancer cells and control of metastases [27,28], we investigate the effects of NB-TA-MSC on the function of NK cells. Herein, we analyze primary young and senescent NB-TA-MSC that have been extensively characterized in a previous paper as TA-MSC, based on the surface phenotype, the differentiation potential typical of BM-MSC and an extended transcriptomic analysis [20]. Our data show that the majority of NB-TA-MSC cells are GD2 positive and display heterogeneous expression of the macrophage checkpoint CD47. Furthermore, all the NB-TA-MSC: (i) Express high levels of the immunoregulatory molecules MHC-I, PD-L1 and PD-L2; (ii) Are resistant to direct cell lysis by activated NK cells and (iii) Inhibit the cytolytic activity of freshly isolated NK through the release of the immunosuppressive molecules L-Kynurenine, a catabolite of indoleamine 2,3-dioxygenase (IDO) and prostaglandin E2 (PGE2). NB-TA-MSC may therefore represent priority targets whose elimination is essential for an improved NB treatment outcome [24,25].

## 2. Materials and Methods

### 2.1. Primary NB-TA-MSC Isolation, Culture and Expansion

NB-TA-MSC (*n* = 6) were derived from NB biopsies and termed as 2ZC, FA, DI, BU, PGE and CO. NB primary cultures were isolated after tumor tissue mechanical dissociation, and collagenase type II treatment as previously described [20]. The cell suspensions were initially collected and plated in polystyrene culture flasks (Corning Costar, Corning, NY, USA) and isolated based on their ability to adhere to plastic as fibroblast-like cells after being in culture at 37  °C in a humidified atmosphere with 5% CO_2_ in D-MEM + GlutaMAX (Gibco, Billings, MT, USA) supplemented with 10% of fetal bovine serum (FBS; Euroclone, Milan, Italy), 50 mg/mL gentamicin and 1% penicillin. Thereafter, NB-TA-MSC were expanded in culture up to senescence at 37 °C in a humidified atmosphere with 5% CO_2_ in the MesenPRO RS™ Medium (Gibco, Billings, MT, USA, 12746012), supplemented with 2 mM L-Glutamine (Gibco, Billings, MT, USA, A29168-01) according to previously described procedures [7]. This medium is specifically formulated to promote the growth of human mesenchymal stem cells (MSCs) for multiple passages while preserving their multi-potential phenotype. In addition, the phenotypic molecular and functional properties of the above-mentioned TA-MSC have been thoroughly characterized elsewhere [20].

### 2.2. Commercial Cell Lines

Human commercial NB cell line SK-N-AS and human erythroleukemia cell line K-562 were purchased from the American Type Culture Collection (ATCC, Manassas, VA, USA) and cultured as already described [27]. Cultures were periodically tested to confirm the absence of mycoplasma by Mycoplasma Detection Kit (Venor-GeM Advance; Sigma Aldrich, St. Louis, MO, USA).

### 2.3. Senescence-Associated Beta-Galactosidase (SA-β-gal) Staining

SA-β-gal staining was performed following the manufacturer’s instructions (BK9860S, Euroclone). Briefly, growth-arrested NB-TA-MSC were washed with PBS, fixed for 15 min at room temperature, incubated in the SA-β-gal staining solution (pH 6.0) overnight at 37 °C, subsequently washed and viewed under a Leica optical microscope [20,29].

### 2.4. Flow-Cytometry

For detection of surface markers, NB-TA-MSC cells were stained with the fluorochrome-conjugated monoclonal antibodies (mAbs) and LIVE/DEAD Fixable Blue Dead Cell Stain Kit (Invitrogen, Thermo Fisher Scientific, Waltham, MA, USA) in PBS 5% FCS for 20 min at 4 °C.

The following mAbs were employed: FITC anti-human MHC-I (HLA A, B, C, #311404, Biolegend), PE-CF594 anti-human PDL-1 (#563742, BD, Franklin Lakes, NJ, USA), Pe-Vio615 anti-human PDL-2 (#130116570, Miltenyi, Bergisch Gladbach, Germany), vioBlue anti-human CD47 (#130-101-359, Miltenyi, Bergisch Gladbach, Germany), AlexaFluor647 anti-human diganglioside GD2 (#562096, BD, Franklin Lakes, NJ, USA), PE anti-human CD105 (#130-112-163, MACS, New York, NY, USA), PE-Cy5 anti-human CD90 (#555597, BD, Franklin Lakes, NJ, USA), PE anti-human CD73 (#550257, BD, Franklin Lakes, NJ, USA), FITC anti-human CD29 (#102206, Biolegend, San Diego, CA, USA) and PE-Cy7 anti-human CD146 (#562135, BD, Franklin Lakes, NJ, USA) were used. A minimum of 5000 events for each condition were acquired with the Beckman-Coulter CytoFLEX LX (Beckman Coulter, Brea, CA, USA). The acquired data were analysed with CytExpert-2.3 (Beckman-Coulter, Brea, CA, USA) and FlowJo v.10 software (BD-Biosciences, Franklin Lakes, NJ, USA). Data were shown as a percentage (%) of the maximum. Otherwise, the data were shown as the fold change of the mean fluorescence intensity ratio (fold change MFI), which represents the ratio between the sample stained with the selected mAb and the control stained with the isotype control antibody.

### 2.5. Quantitative Reverse Transcription PCR (RT-qPCR) Analysis

Total RNA extraction from NB cells was performed with miRNeasy mini kit coupled with on-column DNase-I treatment following the manufacturer’s protocol (Qiagen, Hilden, Germany). 500 ng of total RNA was reversely transcribed with random primers by using Super Script IV first-strand synthesis system following the manufacturer’s instructions (Thermo-Fisher Scientific, Waltham, MA, USA). RT-qPCRs were carried out in triplicate with PowerUp Sybr-Green reagent (Applied-Biosystems, Waltham, MA, USA). GAPDH was used as the endogenous control. RT-qPCRs were run on a QuantStudio-6 Flex instrument (Applied-Biosystems, Waltham, MA, USA). Expression values were calculated by ΔΔCt method using QuantStudio Real-TimePCR-system software v.1.3. The indicated primers were used in the study: WWTR1 (alias TAZ) for 5′-CCTTTTCGCCCAGCACTAGC-3′, rev 5′-ACTCTGTGCCTGGCAGTCTA-3′; NANOG for 5′-ACCAGTCCCAAAGGCAAACA -3′, rev 5′-AGCTGGGTGGAAGAGAACA -3′; OCT4 for 5′-GCCCGAAAGAGAAAGCGAA -3′ rev 5′-AACCACACTCGGACCACATC -3′; SOX2 for 5′-GCCCTGCAGTACAACTCCAT-3′, rev 5′-GACTTGACCACCGAACCCAT-3′; GAPDH for 5′-TCTTTTGCGTCGCCAGCCGA-3′, rev 5′-ACCAGGCGCCCAATACGACC -3′.

### 2.6. Purification of NK Cells

As previously described, fresh NK cells were isolated from peripheral blood mononuclear cells (PBMC) obtained from buffy coats [28]. Buffy coats were collected from volunteer blood donors admitted to the blood transfusion service of OPBG after obtaining informed consent. The Ethical Committee of OPBG approved the study (ID AIRC5x1000 #21147) that was conducted in accordance with the ethical principles stated in the Declaration of Helsinki.

To obtain polyclonal activated NK (aNK) cells, freshly isolated NK cells were cultured on 30 Gy irradiated allogeneic PBMC feeder cells in the presence of 600 U/mL recombinant human IL-2 (Proleukin; Novartis-Farma, Basel, Switzerland) and 1.5 ng/mL phytohemagglutinin (Merck-Millipore, Burlington, MA, USA) for the first week, as previously described [28]. The aNKs were used to perform experiments during the exponential growth phase.

### 2.7. Cytotoxicity Assay

NB-CAF-MSC primary cultures and the K-562 cell line were used as targets of NK cells in a cytotoxicity assay. Target cells were labeled with Cell Tracker Green (CMFDA, Thermo Fisher Scientific, Waltham, MA, USA) according to the manufacturer’s instructions. Ten thousand stained cells were seeded in a V-bottom 96-well plate and then incubated with IL-2-activated NK-cells for 4 h at different effector/target ratios.

Propidium Iodide (PI, Sigma Aldrich, St. Louis, MO, USA) was added to identify dead cells by flow cytometry. Specific lysis was calculated as the percentage of PI^+^ cells among the total CMFDA minus the background level (as determined by target cells incubated without effectors). For the antibody-dependent cellular cytotoxicity (ADCC) assay, we employed the CD105 IgG_1_ mAb TRC105 (Carotuximab, Tracon-Pharmaceuticals Inc., San Diego, CA, USA) NK cells were tested for cytolytic activity against the NB cell lines in a 4 h cytotoxicity assay in the presence of 10 µg/mL of purified anti-CD105 mAb, or IgG mAb (R&D Systems, Minneapolis, MN, USA). The percentage of cell lysis was calculated as indicated elsewhere [27].

### 2.8. Co-Culture of NK and Target Cells

In co-culture experiments with NB-TA-MSC cells, freshly isolated NK cells were added to NB-TA-MSC cells in direct cell contact or separated by 0.4 μM pore size transwell chambers (NB-TA-MSC cells to NK cell ratio of 1:5) in RPMI medium, supplemented with 600 U/mL of IL-2. After 6 days of co-culture, the NKs were collected and analyzed.

Co-culture experiments were performed either in the absence or presence of IDO and PGE2 synthesis inhibitors. We used: 1 mM of 1-Methyl-D-Tryptophane (1-MT, Sigma-Aldrich, St. Louis, MO, USA) as an IDO inhibitor and 5 µM N-[2-(Cyclohexyloxy)-4-nitrophenyl]methanesulfonamide (NS-398, Sigma-Aldrich, St. Louis, MO, USA) as an inhibitor of PGE2 synthesis. After 6 days, NK cells were collected and used as effectors in the cytotoxicity assays. The K-562 cell line was a control target for NK cell functional assays.

NK Proliferation was evaluated by electronic counting at the end of the co-culture experiments. The seeded sample represents the number of cells plated before co-culture (3 × 10^5^ cells). PI- NK cells were acquired with the Beckman-Coulter CytoFLEX-LX flow-cytometer. A minimum of 5000 events for each condition were acquired, and data were analysed with CytExpert-2.3.

### 2.9. Statistical Analysis

Statistical analysis was performed with GraphPad Prism 8 Software (GraphPad Software 8.0.1). As indicated, statistical significance was calculated using a two-tailed paired Student’s t-test with Bonferroni correction for multiple comparisons. Data were expressed as mean ± standard deviation (SD); *p* values < 0.05 were considered statistically significant.

## 3. Results

### 3.1. NB-TA-MSC Display MSC Phenotype and a Heterogeneous Expression of Immunomodulatory Molecules

We analyzed young NB-TA-MSC for the expression of MSC and immunomodulatory markers. We used TA-MSC derived from four NB biopsies previously characterized [20], and we termed these cells 2ZC, FA, BU and DI. Flow cytometry analysis confirmed that the majority of the primary cultures express both surface markers (Appendix A) and transcripts of stemness markers (Appendix A) typical of MSC [20].

Interestingly, all the tested NB-TA-MSCs express the checkpoint ligands PD-L1 and PD-L2. The macrophage checkpoint CD47, delivering the “do not eat me” signal [30], is expressed by DI and BU but not by 2ZC and FA. GD2, is a marker typical of neoplastic neuroblasts that is a target for immunotherapy protocols [31]. It is expressed by NB-TMSC with the exception of FA (Figure 1A). We also found that NB-TA-MSCs express the mesenchymal immunomodulatory transcriptional coactivator WWTR1 (alias TAZ; Figure 1B). Finally, these cells also display high levels of the immunomodulatory molecule MHC-1.

Replicative senescence (growth-arrest) of normal MSC is characterized by some functional defects [20,24,32,33], and senescent cells are identified for the SA-β-gal staining. Analysis of Figure 2A shows that young proliferating NB-TA-MSC are negative for the SA-β-gal staining, while a prolonged culture of 2ZC primary cell culture produced non-proliferating culture with ≥95% of cells positive for the SA-β-gal staining (Figure 2B), indicating that almost all of these cells become senescent at high-passage. We also received two other high-passage NB-TA-MSC primary cultures, CO and PGE cells, which underwent senescence very rapidly (Figure 2B). Thus, these cells were characterized only for their interactions with NK cells.

### 3.2. NB-TA-MSC Primary Cell Cultures Are Resistant to NK Cell-Mediated Lysis

Subsequently, we investigated whether NB-TA-MSC cells interfered with the cytolytic potential of NK cells. First, we analyzed the cytolytic activity of activated NK (aNK) cells against four NB-TA-MSC primary cultures in 4 h co-culture assays. As shown in Figure 3A, low passage proliferating NB-TA-MSC cell lines are highly resistant to aNK cell lysis, while senescent PGE (Figure 3B) are highly sensitive to aNK cell lysis. The aNK-mediated cell lysis of proliferating NB-TA-MSC primary cultures is improved in the presence of the CD105 IgG1 monoclonal antibody TRC105, which activates ADCC (Figure 3).

### 3.3. NB-TA-MSC Primary Cultures Inhibit the Cytolytic Activity of NK Cells

Next, we analyzed whether NB-TA-MSC interfered with the proliferation and the cytolytic potential of NK cells. For this purpose, freshly isolated NK cells and NB-TA-MSC cells were co-cultured in the presence of IL-2 under direct cell–cell contact or under transwell conditions, as illustrated in Figure 4A. On day 6, NK cells were collected and tested for their cytolytic activity against K-562 and their proliferation. All NB-TA-MSC primary cultures (i) Strongly inhibit the cytolytic activity of co-cultured NK cells against K-562 target cells (Figure 4B) and significantly affect their proliferation (Figure 5A), either in conditions of direct (cell–cell contact) or indirect contact (transwell conditions).

### 3.4. NB-TA-MSC Primary Cultures Down-Regulate the Expression of Activating Receptors on NK Cells

We analyzed whether the inhibitory effect of NB-TA-MSC on co-cultured NK cells reflected the downregulation of activating NK receptors. All the NB-TA-MSC primary cultures induce a decreased expression of NKG2D and natural cytotoxicity receptors (NCRs) NKp30 and NKp44, as well as CD69 (Figure 5B), either in conditions of direct or indirect contact. In addition, the co-culture of NB-TA-MSC-DI and BU also caused a decreased expression of NKp46 in NK cells compared to the control.

### 3.5. Compounds Inhibiting the Soluble Immune Regulatory Molecules Kynurenine and PGE2 Preserve Nk Cells Cytolytic Activity and Proliferation

The impairment of NK cytolytic activity and proliferation under cell–cell contact is partially prevented in the presence of compounds targeting the IDO catabolite kynurenine and PGE2, which are frequently released by tumor cells (Figure 6A,B). This data suggests that the release of soluble factors by NB-TA-MSC is a mechanism involved in NK cell immunosuppression.

### 3.6. Senescent NB-TA-MSC Lose Their Immunomodulatory Properties

Subsequently, we tested whether senescent NB-TA-MSC could maintain their immunoregulatory properties on NK cells. NK cells co-cultured with growth-arrested senescent PGE, 2ZC and CO maintained efficient cytotoxic activity against K-562 cells (Figure 7A,B). Moreover, we compared young/proliferating and senescent NB-TA-MSC-2ZC for their capacity to affect the cytotoxic potential of NK cells against K-562 target cells at different E:T ratios, based on direct or indirect cell contact (Appendix A). Whereas young NB-TA-MSC-2ZC efficiently impaired the cytotoxic potential of NK cells, senescent NB-TA-MSC-2ZC did not.

Finally, we analyzed the effect of senescent NB-TA-MSC on the proliferation potential of NK cells, expressed as the number of live NK cells (Figure 7C). Co-culture of NK cells with senescent NB-TA-MSC does not alter the proliferation of NK cells, with the exception of the co-culture with NB-TA-CO that causes a significant decrease of NK cell number, but only under direct cell–cell contact.

## 4. Discussion

In many solid tumors, cancer-associated stromal cells enhance cancer cell proliferation, angiogenesis, invasion and metastases, especially in the context of chronic inflammation interacting preferentially with tumour-associated macrophages (TAMs) [34]. In addition, these cells display immunoregulatory properties through the production of PGE2 [21]. In NB, a subset of αFAP and FSP-1 expressing CAFs share characteristics and pro-tumorigenic activity with mesenchymal stromal cells, therefore they have been termed CAF-MSCs [19,20], and their pro-tumorigenic potential in NB progression has been confirmed by different groups [8,14,19,20,21,22,23,24,25]. Moreover, intratumoral TAMs/T-MSCs density significantly correlated with clinical stage, MYCN amplification, bone marrow metastases, histological classification, histological type, and risk classification [23], while effects on NK cells have not been demonstrated in the above-mentioned papers.

Therefore, herein we further analyzed four proliferating NB-TA-MSCs primary cultures for their immunoregulatory properties on purified and activated NK cells [20]. We characterized their phenotype concerning the expression of molecules displaying immunomodulatory functions on NK cells, as well as their interactions with NK cells. Flow cytometry analysis shows that these NB-TA-MSCs exhibit a heterogeneous phenotype. Indeed, even if all of them express high levels of the MSC markers CD105, CD73, CD90, CD29 and CD146, as well as of the immunomodulatory molecules MHC-I, PD-L1 and PD-L2, one of them (NB-TA-MSC-FA) does not express the macrophage checkpoint CD47 and GD2, a marker of neoplastic neuroblasts that constitutes the main target for different immunotherapy protocols [31]. RT-qPCR analysis shows that all the NB-TA-MSC express high levels of the TAZ transcript that displays important immunoregulatory properties on T cells and NK cells [27] as well as the transcripts for the stemness markers NANOG, OCT4 and SOX2.

Subsequently, we studied the sensitivity of NB-TA-MSC to activated NK cell-mediated killing. Low passage proliferating NB-TA-MSC resulted in high resistance to NK-mediated direct lysis. However, this resistance was overcome by utilizing the CD105 mAb TCR105 through the activation of ADCC. The resistance of NB-TA-MSC to aNK cell-mediated killing is an important property that distinguishes them from normal BM-derived MSC that have been reported to be highly sensitive [35] and further highlights the “pro-tumorigenic nature” of TA-MSCs.

It has been reported that normal MSCs undergoing replicative senescence do not exhibit phenotypic modifications, their susceptibility to NK-cell lysis remains stable, and they inhibit T cells less efficiently, but not NK cell proliferation [32,33]. Nevertheless, we found that senescent NB-TA-MSC, behaving differently from normal senescent MSCs, appeared almost totally sensitive to activated NK-mediated direct lysis, probably because of their decreased expression of MHC-I, PDL-1 and PDL-2.

Moreover, young, but not senescent, NB-TA-MSCs, upon co-culture, efficiently inhibit the function of NK cells, which display poor cytotoxic activity against target K-562 cells and a significantly decreased surface expression of different activating molecules such as NKG2D, NKp30, NKp44 and CD69. Concerning the inhibition of NK proliferation, senescent NB-TA-MSC exhibits a heterogeneous behavior, since senescent 2ZC is still able to counteract the growth of co-cultured NK cells, while senescent PGEs have lost this property.

The inhibition of NK cell proliferation is a major event that probably correlates with the low intratumoral NK density, detected in high-risk patients with MYCN amplification status and associated with a poor prognosis [18]. These immunomodulatory effects are mediated both by cell–cell contacts and by soluble factors. Concerning the inhibition through cell–cell contact, it has been reported that CAF-MSC and macrophages influence each other by cell–cell interactions increasing their pro-tumorigenic potential [22,23] reciprocally. Thus, it is likely that NB-TA-MSC may also interact with NK cells through a cell–cell contact mechanism that could be controlled by the co-transcriptional factor TAZ, strongly expressed in TA-MSC. In this respect, we have recently reported that NB cell lines, expressing phenotype and functions typical of BM-MSC (NB-MSC CD105+/TAZ+), inhibit the cytotoxic functions of NK cells exclusively by cell–cell contact mechanisms and that TAZ-silencing rescues these properties [27]. Our data suggest that the impairment of cytolytic activity and proliferation of NK cells in conditions of cell–cell contact, partially restored in the presence of specific inhibitors for L-kynurenine and PGE2, is due either to the release of soluble immunosuppressive factors [22,23] or to mechanisms not involving the release of soluble molecules [32]. In this respect, it has been shown that MSC-derived exosomes inhibit proliferation, activation, and cytotoxicity of NK cells via TGFβ-dependent production of L-kynurenine and PGE2 [36]. Thus, the release by NB-TA-MSC of immunomodulatory exosomes could represent another mechanism by which these cells inhibit NK cell function.

The release of L-kynurenine and PGE2 in the NB tumor microenvironment, either as soluble factors or conveyed by exosome [35], is not surprising since NB-TA-MSC are the main producers of PGE2 [21] and L-kynurenine is produced in the neuroblastoma TME with major immunosuppressive functions [4].

Even if NB-TA-MSCs are phenotypically and functionally similar to BM-MSC [19,20], transcriptomic profiling results identified about 500 genes differentially expressed [20,37]. In NB-TA-MSC, genes related to the Wnt signaling pathway, stemness and epithelial-to-mesenchymal transition program are up-regulated, and genes related to the immune response are down-regulated. They could be strongly correlated to the immunoregulatory functions of NB-TA-MSC against T [20] and NK cells.

In human NB, GD2 is expressed essentially on neuroblasts, and monoclonal antibodies or CAR T-cells targeting GD2 have substantially improved outcomes for children with high-risk neuroblastoma [38,39]. However, approximately 40% of patients with NB still relapse after anti-GD2 treatments due to interferences by different elements present in the NB-TME [4,19,21,23,24,40,41]. Since also NB-TA-MSC may express GD2, owing to their important immunosuppressive properties, these cells could constitute a TME component refractory to anti-GD2 treatments. Thus, we suggest that the treatment of patients with high-risk NB with anti-GD2 protocols may be improved by either eliminating or disarming TA-MSCs, choosing alternative surface targets such as CD105 [24] or targeting particular molecules or pathways that mediate their pro-tumorigenic and immune suppressive effects. In this context, senescent NB-TA-MSCs lose the expression of immunomodulatory molecules, as well as the important immunosuppressive properties found in young NB-TA-MSCs. Thus, in vivo induction of senescence in NB-TA-MSCs would disarm these cells. Senescence of MSC is induced by elevated ROS levels [41] that can be obtained with anti-cancer drugs such as isoalantolactone active both in vitro and in vivo [42].

## 5. Conclusions

In conclusion, NB-TA-MSC displays important immunosuppressive properties towards NK cells, affecting NK cell cytotoxicity and proliferation. Therefore, NB-TA-MSC may constitute a priority therapeutic target in the TME, and their elimination could be necessary for the improvement of immunotherapy protocols.

## Figures and Tables

**Figure 1 cancers-15-00019-f001:**
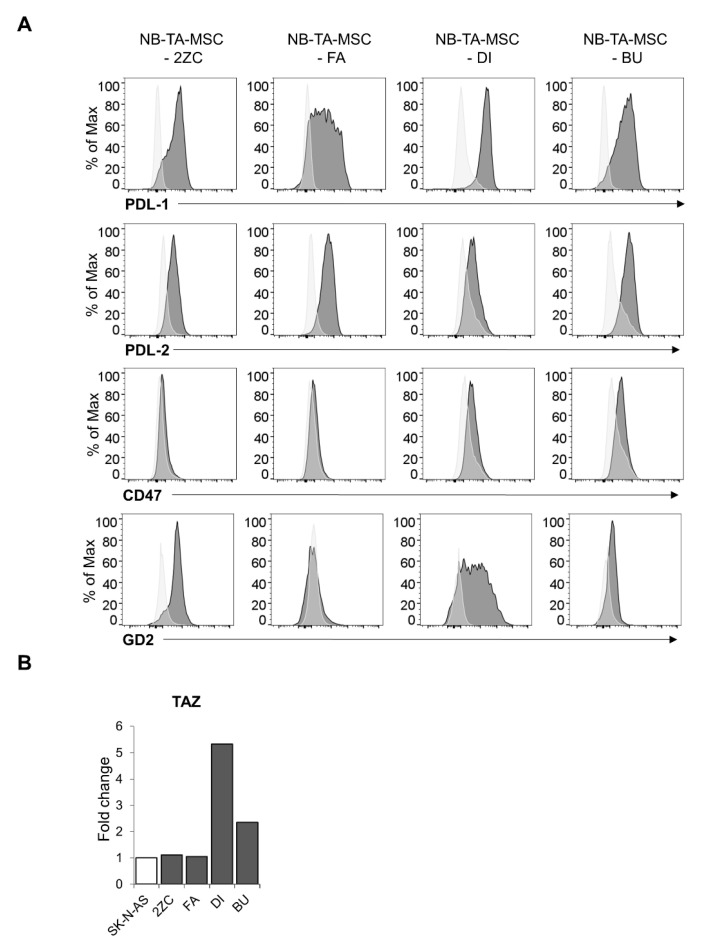
Evaluation of the expression of immunomodulatory molecules in primary NB-TA-MSC cultures. (**A**) Flow cytometry analysis of the indicated surface markers (PD-L1, PD-L2, CD47 and GD2) in primary NB-TA-MSC cultures. Light grey histograms represent unstained control; dark grey histograms represent stained samples. A representative experiment is shown of *n* = 5 experiments performed. (**B**) RT-qPCR analysis of TAZ transcript in different primary NB-TA-MSC cultures. The NB commercial cell line SK-N-AS was used as a reference control since these cells display a mesenchymal phenotype and high TAZ expression [27]. Histograms represent the fold change of gene transcript expression normalized for GAPDH expression compared to SK-N-AS expression, whose level is arbitrarily set as 1. Data are expressed as mean ± SD (*n* = 3).

**Figure 2 cancers-15-00019-f002:**
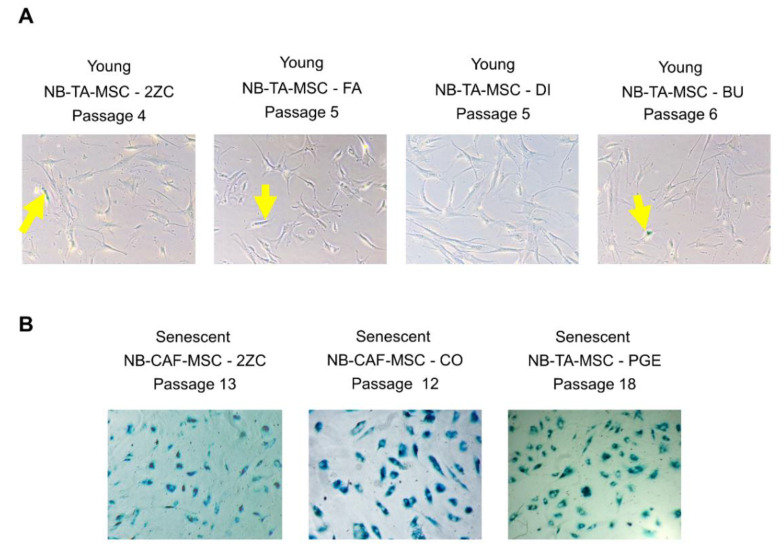
Staining with the senescence-associated beta-galactosidase (SA-β-gal) assay of young/proliferative and growth-arrested NB-TA-MSC. (**A**) Representative images of young/proliferating NB-TA-MSC and (**B**) Growth-arrested NB-TA-MSC stained with SA-β-gal at the indicated culture passages. The senescent cells appear stained in blue. Images represent 10× with phase contrast optical microscope of *n* = 3 independent experiments. Yellow arrows indicate rare β-gal+ cells in young proliferating NB-TA-MSC cells.

**Figure 3 cancers-15-00019-f003:**
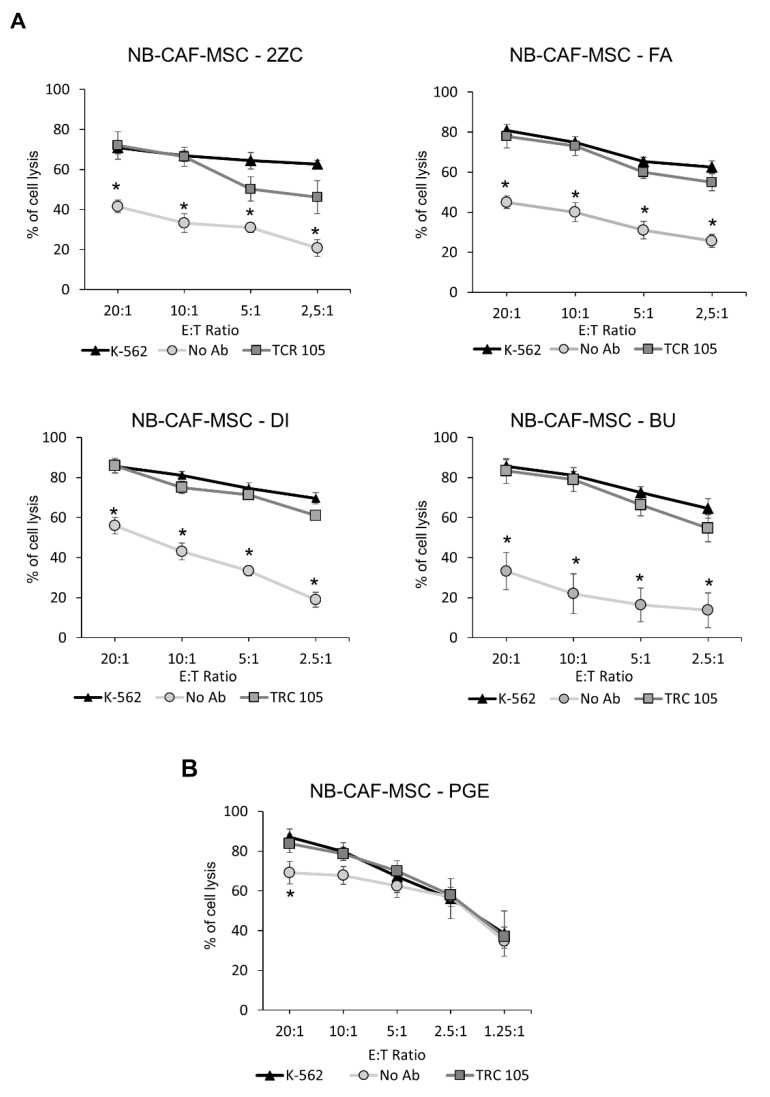
Susceptibility of primary NB-TA-MSC cultures to aNK cell-mediated lysis. (**A**) Allogeneic aNK cells were used as effector cells against CMFDA-labelled young/proliferating and (**B**) Senescent NB-TA-MSC-PGE primary cultures. Target cells were labelled with CMFDA, and allogeneic aNK cells were used as effectors at different E:T ratios as indicated. CMFDA-labelled K-562 cells were used as a positive control of lysis (K-562). An anti-CD105 IgG mAb, that induced ADCC (TRC 105) or irrelevant IgG mAb as a control (No Ab) was used. Data were expressed as mean ± SD (*n* = 4) of the percentage of cell lysis (CMFDA+ and PI+ cells). * *p* < 0.05 No Ab vs. K-562.

**Figure 4 cancers-15-00019-f004:**
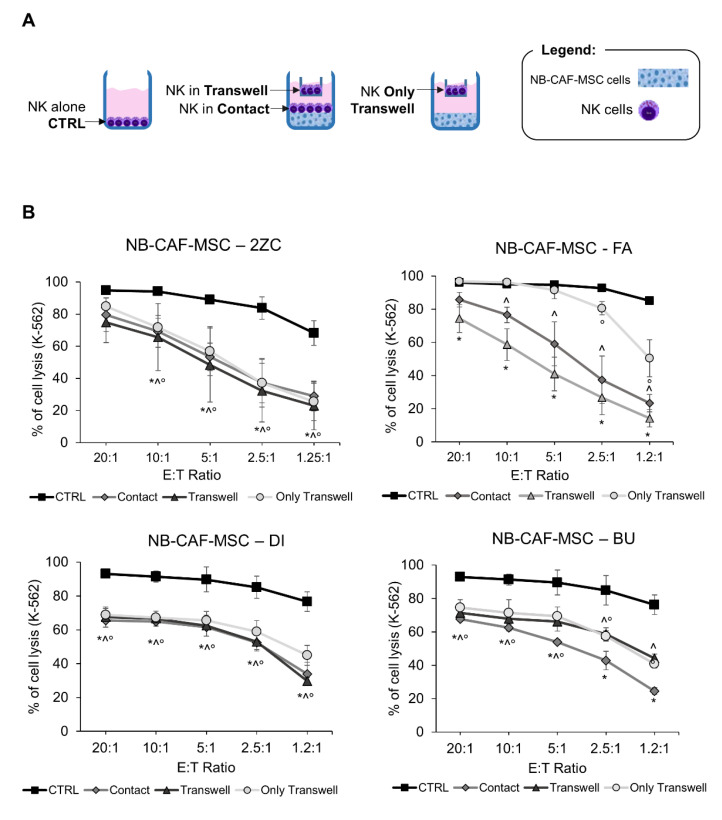
Evaluation of NK cell cytotoxic activity after co-culture with primary NB-TA-MSC. (**A**) Schematic representation of cells seeded in co-culture experiments. Freshly isolated NK cells were cultured in the upper chamber for 6 days with NB-TA-MSC cells in the lower chamber (Only Transwell) or a setting with NK cells in the upper chamber (NK Transwell) and NB-TA-MSC with NK cells in the lower chamber (Contact). NK cultured alone were used as the control (CTRL). (**B**) NK cell cytotoxicity assays against CMFDA-labelled K-562 cells after co-culture with young/proliferating NB-TA-2ZC, FA, DI and BU primary cultures under direct cell–cell contact (Contact) or under Transwell conditions (Transwell and Only Transwell). NK cultured alone were used as a control (CTRL). Percentages of cell lysis (CMFDA+ and PI+ cells) were expressed as mean ±SD (*n* = 4). * *p* < 0.05 Contact vs. CTRL. ^ *p* < 0.05 Transwell vs. CTRL ° *p* < 0.05 Only Transwell vs. CTRL.

**Figure 5 cancers-15-00019-f005:**
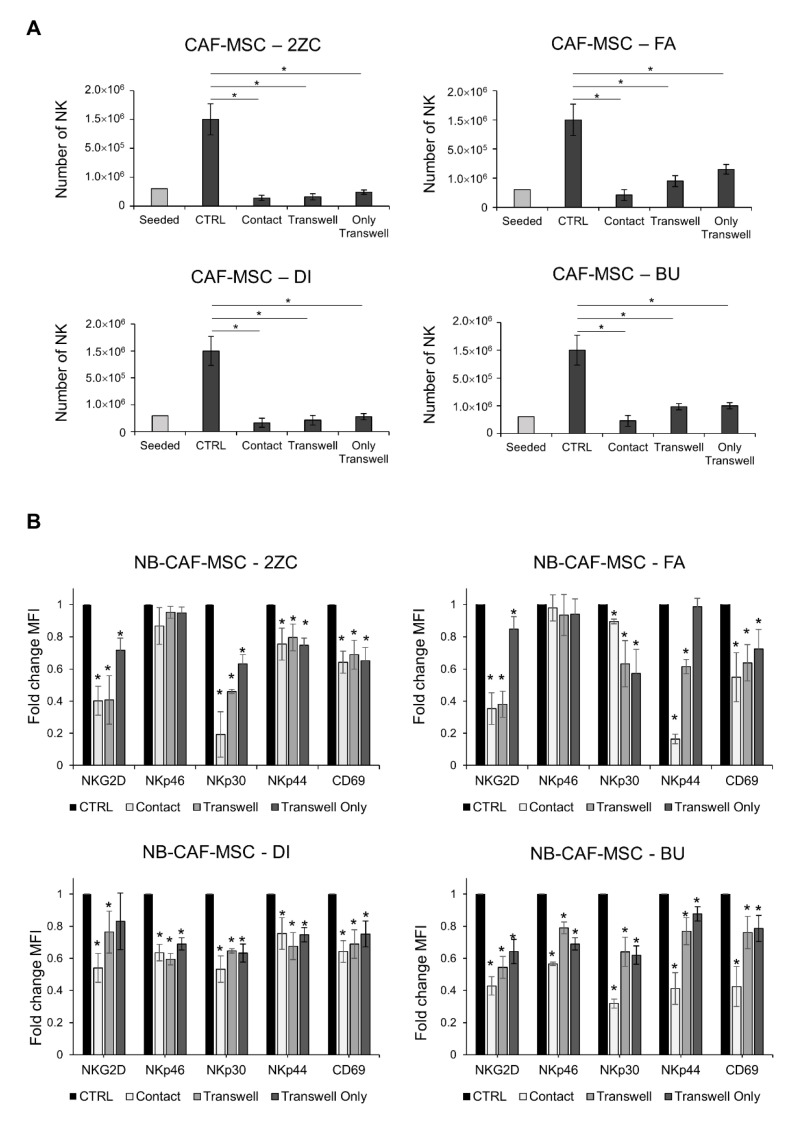
Evaluation of NK cell proliferative potential and downregulation of NK activating receptors after co-culture with primary NB-TA-MSC cultures. (**A**) Live NK cell number (PI^−^) after co-culture with NB-TA–MSC primary cultures. Data were expressed as mean ±SD (*n* = 6). * *p* < 0.05 vs. CTRL. (**B**) Flow cytometry analysis of the activating receptors present on NK cells after co-culture with the indicated young/proliferating NB-T-MSC primary cultures under direct cell–cell contact or under Transwell conditions. Data were expressed as Fold change MFI compared with CTRL ± SD (*n* = 3). * *p* < 0.05 vs. CTRL.

**Figure 6 cancers-15-00019-f006:**
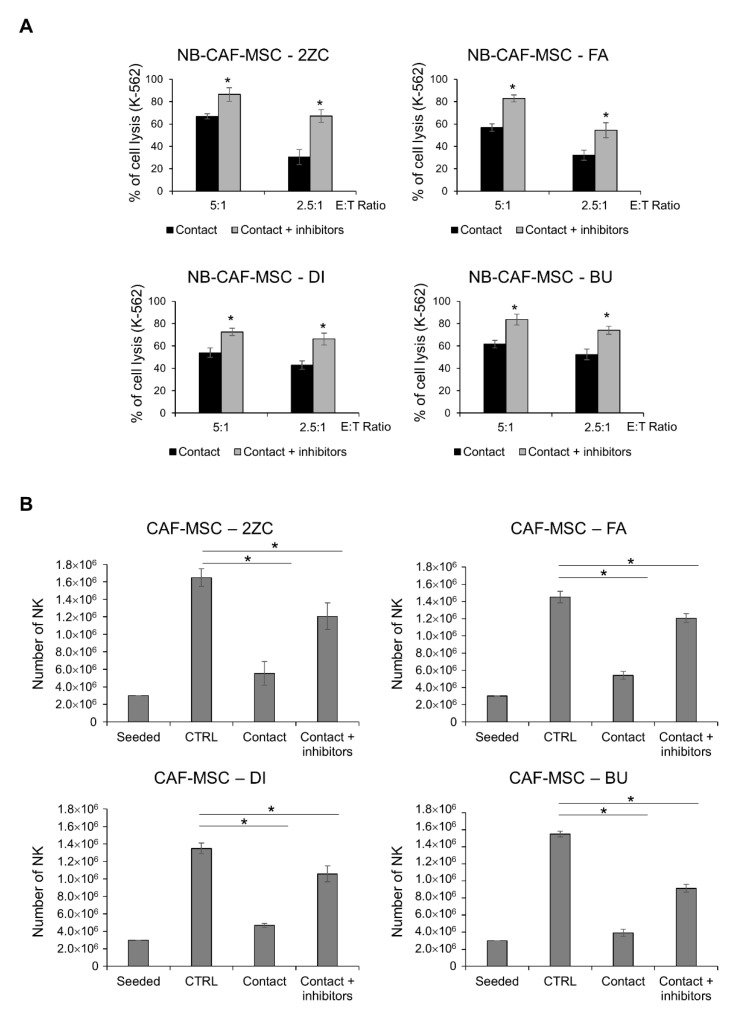
Effect of IDO and PGE2 inhibitors on NK cytotoxicity and proliferation under cell-cell contact conditions. (**A**) Percentage of K-562 cells lysis in cytotoxicity assays using freshly isolated NK cells after co-culture for 6 days with young/proliferating TA–MSC 2ZC, FA, DI and BU primary cultures either in the presence or in the absence of IDO and PGE2 inhibitors (Contact and Contact + inhibitors). Values are expressed as mean ± SD (*n* = 3). * *p* < 0.05 vs. NK Ctrl and vs. NK Ctrl + IDO and PGE2 inhibitors. (**B**) Live NK cell number (PI^−^) after co-culture with NB-TA–MSC primary cultures, either in the presence or in the absence of IDO and PGE2 inhibitors (Contact and Contact + inhibitors). Starting number of seeded NK cells and NK cultured alone (CTRL) were used as controls. Data were expressed as mean ± SD (*n* = 6). * *p* < 0.05 vs. CTRL.

**Figure 7 cancers-15-00019-f007:**
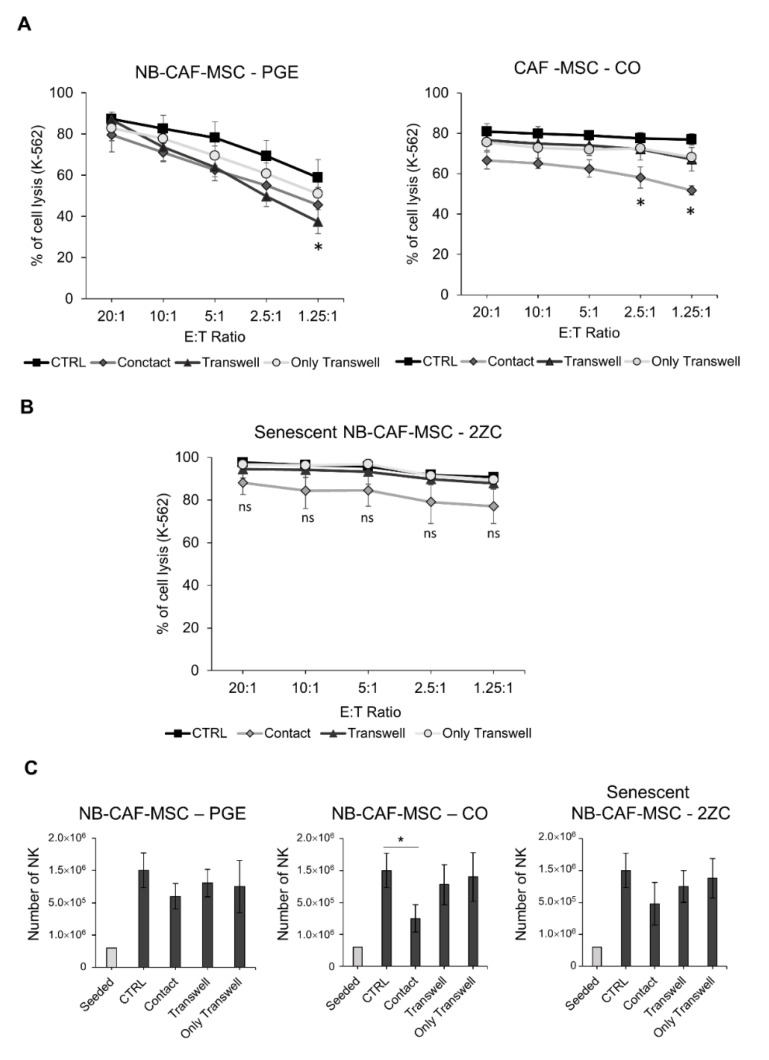
Evaluation of NK cell cytotoxic activity and proliferative potential after co-culture with senescent NB-TA-MSC cultures. NK-cell cytotoxicity assays against CMFDA-labelled K-562 cells after co-culture with (**A**) Senescent NB-TA–MSC-CO, PGE (**B**) Senescent NB-TA–MSC-2ZC culture, under direct cell-cell Contact or under Transwell conditions. Percentages of lysed cells were expressed as mean ± SD (*n* = 3). ns = not significant. (**C**) Live NK cell number (PI^−^) after co-culture with senescent NB-TA–MSC cultures. Co-culture of senescent NB-TA-MSC-PGE and 2ZC with NK cells did not affect cell number compared to CTRL, while the co-culture with senescent NB-TA-MSC-CO in direct-contact condition caused a slowdown of NK cells number. Data were expressed as mean ± SD (*n* = 6). * *p* < 0.05 vs. CTRL.

## Data Availability

The data presented in this study are available within the article and its supplementary data files.

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
