# Peer review of "Neuroblastoma Tumor-Associated Mesenchymal Stromal Cells Regulate the Cytolytic Functions of NK Cells"

_cancers, 2022, doi:10.3390/cancers15010019_

Round 1

Reviewer 1 Report

In this manuscript, Di Matteo et al. studied the immune-regulatory potential of six primary young and senescent NB-TA-MSC on NK cell function. They showed that young cells display a phenotype (CD105+/CD90+/CD73+/CD29+/CD146+) typical of MSC cells and also express high levels of immunomodulatory molecules (MHC-I, PDL-1 and PDL-2 and 37 transcriptional-co-activator WWTR1) able to hinder NK cell activity. They also showed that four of them express the neuroblastoma marker GD2, the most common target for NB immunotherapy. young NB-TA-MSC, contrary to the senescent ones, are shown to be resistant to activated NK cell-mediated lysis, but this behavior is overcome using anti-CD105 antibody TRC105 that activates antibody-dependent cell-mediated cytotoxicity. In addition, proliferating NB-TA-MSC, but not senescent ones are shown to inhibit, after 6 days of co-culture, proliferation, expression of activating receptors and cytolytic activity of freshly isolated NK, but inhibitors of the soluble immunosuppressive factors L- kynurenine and prostaglandin E2 efficiently counteract this latter effect. Their data indicate the presence of phenotypically heterogeneous NB-TA-MSC displaying potent immuno-regulatory properties towards NK cells, whose inhibition could be mandatory in order to improve antitumor efficacy of targeted immunotherapy.

Experiments are well-designed and comprehensive. Obtained results are in concordance with discussions. There are other papers indicating the importance of inhibitory effect of MSCs on NK cells (1,2) which points out the importance of this interaction in terms of different cancers that are difficult to treat such as neuroblastoma. In this context, their study results and future prospects/recommendations are very promising for choosing the targets for a more efficient therapy.  

1-Abbasi B, Shamsasenjan K, Ahmadi M, Beheshti SA, Saleh M. Mesenchymal stem cells and natural killer cells interaction mechanisms and potential clinical applications. Stem Cell Res Ther. 2022 Mar 7;13(1):97. doi: 10.1186/s13287-022-02777-4. PMID: 35255980; PMCID: PMC8900412.

2-Spaggiari GM, Capobianco A, Abdelrazik H, Becchetti F, Mingari MC, Moretta L. Mesenchymal stem cells inhibit natural killer-cell proliferation, cytotoxicity, and cytokine production: role of indoleamine 2,3-dioxygenase and prostaglandin E2. Blood. 2008 Feb 1;111(3):1327-33. doi: 10.1182/blood-2007-02-074997. Epub 2007 Oct 19. PMID: 17951526.

Author Response

We are pleased that the editorial team and Reviewers found merits in our study. All changes have been marked in red in the revised manuscript.

Reviewer 1

We sincerely thank the Reviewer for appreciating our work. The Reviewer indicates two other interesting publications on interactions between NK cells and MSCs. We cited Spaggiari et al in our manuscript, but we had not included Abbasi B et al. Now, we also added this reference in Introduction of the revised manuscript.

Reviewer 2 Report

The authors provide a robust overview and a sufficent context of their research. However, a more "short-cut-version" of the expression profiles of their targeted cells (Fig. 1 and 2) would make the manuscript more easy to read (one could still put the flow-cytometry analysis in the supplement). Moreover, a graphical abstract of the main hypothisis would be nice but is not mandatory. 

Taken together I would like to congratulate the author for a very important work in the context of a very devastating disease. 

Author Response

We are pleased that the editorial team and Reviewers found merits in our study. We have attempted to address all the issues raised by the Reviewer.  All changes have been marked in red in the revised manuscript.

Reviewer 2

We sincerely thank the Reviewer for appreciating our work and for her/his useful comments.

As suggested, we created a new “short-cut version” of Figures 1 and 2. In particular, we moved flow cytometry and real time PCR for mesenchymal stem cell markers in Supplementary figure 1, since they are mainly confirmatory of the MSC phenotype. Now, the new main Figure 1 includes the immunomodulatory surface markers (Figure 1A) and TAZ expression (Figure 1B). To simplify this figure, we also decided to remove MHC I expression (mentioned as “data not shown”).

Further, we agree with the Reviewer’s suggestion and we created a graphical abstract summarizing our work, enclosed in the revised version of the manuscript.